# A Cooperative Target Localization Method Based on UAV Aerial Images

Minglei Du [1,2,*], Haodong Zou [3], Tinghui Wang [1,4] and Ke Zhu [2]

1    School of Astronautics, Northwestern Polytechnical University, Xi'an 710072, China;
     wth_nwpu@mail.nwpu.edu.cn
2    Xi'an Institute of Modern Control Technology, Xi'an 710065, China; 21925035@zju.edu.cn
3    Unmanned System Research Institute, Northwestern Polytechnical University, Xi'an 710072, China;
     zouhaodong@mail.nwpu.edu.cn
4    Shaanxi Key Laboratory of Aerospace Vehicle Design, Xi'an 710072, China
*    Correspondence: mldu@mail.nwpu.edu.cn

**Abstract:** A passive localization algorithm based on UAV aerial images and Angle of Arrival (AOA) is proposed to solve the target passive localization problem. In this paper, the images are captured using fixed-focus shooting. A target localization factor is defined to eliminate the effect of focal length and simplify calculations. To synchronize the positions of multiple UAVs, a dynamic navigation coordinate system is defined with the leader at its center. The target positioning factor is calculated based on image information and azimuth elements within the UAV photoelectric reconnaissance device. The covariance equation is used to derive AOA, which is then used to obtain the target coordinate value by solving the joint UAV swarm positional information. The accuracy of the positioning algorithm is verified by actual aerial images. Based on this, an error model is established, the calculation method of the co-localization PDOP is given, and the correctness of the error model is verified through the simulation of the Monte Carlo statistical method. At the end of the article, the trackless Kalman filter algorithm is designed to improve positioning accuracy, and the simulation analysis is performed on the stationary and moving states of the target. The experimental results show that the algorithm can significantly improve the target positioning accuracy and ensure stable tracking of the target.

**Keywords:** UAVs; co-localization; error analysis; cubature Kalman filter

## 1. Introduction

Reconnaissance-type UAVs are equipped with key features that enable them to locate targets quickly and accurately, as well as predict their behavior with precision. As technology has advanced, UAVs have become capable of multi-machine collaborative operations, thanks to bionic clustering and communication networking technologies. Optoelectronic information technology has also undergone significant development, resulting in the integration, miniaturization, and cost-effectiveness of airborne optoelectronic detection devices [1,2]. To further improve target localization accuracy, UAVs now employ a clustered approach to execute target localization and situational awareness duties [3].

In general, there are two types of UAV target localization techniques: active localization and passive localization. Active localization is the process of actively locating a target using a radio instrument, such as a UAV radar. The UAV actively ranges the target while actively positioning itself, which has a bigger impact on the UAV's own concealing abilities and survivability [4,5]. By passively collecting target information rather than actively producing electromagnetic waves, lasers, etc., to obtain ranging information, passive placement helps to some extent, ensuring the safety of the UAV itself. According to the type of observation quantity, passive localization techniques are divided into several categories: primarily Collinear Equation, Image Matching, Binocular Vision 3D Localization, Doppler Rate of

Frequency Change (DRC), Doppler Rate of Chang (DRC), Phase Difference Rate of Change (PDRC), Time Difference of Arrival (TDOA), Frequency Difference of Arrival (FDOA), Angle of Arrival (AOA), and other techniques [6,7]. The UAVs mentioned in this paper perform clustered localization tasks, and they distinguish themselves by being small, light, and having low power consumption, as well as better anti-jamming and stealthiness. To accommodate the UAV platform and usage needs, localization techniques need to be improved.

Collinear equation, image matching, and binocular vision 3D localization methods in the aforementioned passive localization are localization methods based on image information that can localize the target via a single image but with significant localization error. The flat terrain assumption, which is not always true in real-world application circumstances, is the foundation of the covariance equation approach. Although feature-based image matching is more efficient and gray-scale correlation-based image matching is more widely used, image-matching algorithms are more difficult to use, take longer to complete, and demand more computing resources, and thus they cannot be employed in situations where real-time performance is crucial. The secret to binocular or multicamera vision 3D localization is to shoot the target from various angles and acquire local feature points of the object, which cannot satisfy the measurement accuracy of further away targets due to the restriction of baseline distance. Although direction-finding cross-localization improves target maneuvering performance prediction, it has a significant flaw in multi-target localization and falls short of UAVs' general criteria. In wireless sensor networks, where target information is typically received from sensors mounted on several observation points, methods like DRC, PDRC, TDOA, FDOA, and AOA are based on fast-improving localization algorithms [8].

To sum up, this work suggests an enhanced passive localization technique with the following key contributions based on the picture data obtained from aerial photography.

1.　The solution technique does not require the input of focal length and elevation information;
2.　Simultaneous localization of multiple targets is possible;
3.　The target localization error may be estimated based on the error component of each observation;
4.　The proposed traceless Kalman filtering approach can significantly increase the target localization and tracking accuracy while maintaining good robustness.

The article is organized as follows. Section 2 discusses the multi-UAV cooperative target localization method, including algorithm assumptions and a schematic depiction of the computational flow. Section 3 introduces the multi-UAV target cooperative localization algorithm. Section 4 examines the localization error and constructs a cooperative localization error model based on Section 3. Section 5 describes the traceless Kalman filter's principles and computational methods. Section 6 simulates the algorithm's correctness and highlights its benefits and drawbacks. Section 7 presents the conclusions.

## 2. Scenario Problem Description

### 2.1. Cooperative Target Localization Process for Multiple UAVs

The scenario is described in terms of multiple UAVs performing real-time reconnaissance and localization missions, as follows: The UAV is equipped with an electro-optical load to obtain wide-field of view, high-resolution infrared and visible image information, allowing both target and target-assisted localization to be performed.

Through mission mustering, multiple UAVs in the scenario area coordinate to pinpoint the objective. Within the electric-optic load action range, multiple UAVs gather the corresponding target pixel coordinates based on image data, sync the image data with the appropriate navigation data to determine the target's relative position using the pertinent interior orientation data, and then convert the target's absolute position data, The specific process is shown in Figure 1. A single UAV's positioning process requires preassembled elevation or range information. We design a collaborative target placement solution approach in the absence of elevation information, taking into account the benefits of passive positioning and relative height measurement accuracy. Continuous tracking and gazing of

the target is impossible due to the complex combat environment, but to take advantage of the UAV's wide field of view for efficient reconnaissance in a limited time window, it is necessary to complete multiple target localization solutions based on multiple images. Furthermore, because absolute target position information is required, multiple UAVs should establish spatial relative relationships through position sharing prior to collaborative target localization, and the time uniformity problem is solved by synchronizing and fusing multiple information of respective UAVs.

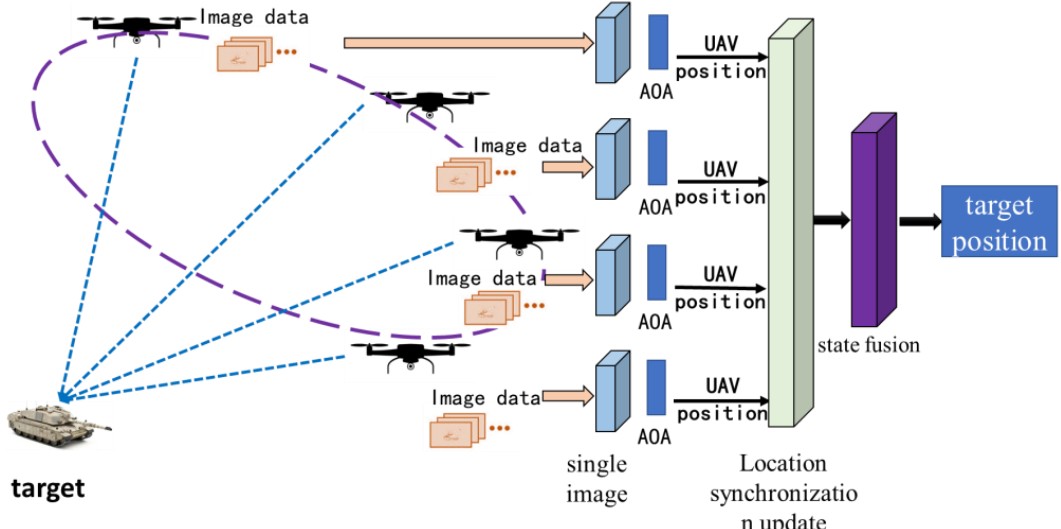

**Figure 1.** Schematic diagram of the process of Multi-UAV performing target positioning tasks.

*2.2. Model Assumptions*

The following assumptions are made in the above scenario problem:

(1) Because the UAV's camera center corresponds with the origin of the navigation coordinate system, any position mistake between them is ignored.
(2) The UAV's own location information is updated without delay;
(3) The data link has no latency, a big bandwidth, and anti-interference properties to ensure that information is properly transferred.
(4) The image's optical distortion is ignored.

The input parameters for cooperative target localization of numerous UAVs are primarily separated into the following categories: UAV flight status parameters, navigation data, and picture data, among others. The output parameter is the location information of the target, and the specific calculation process is shown in Figure 2.

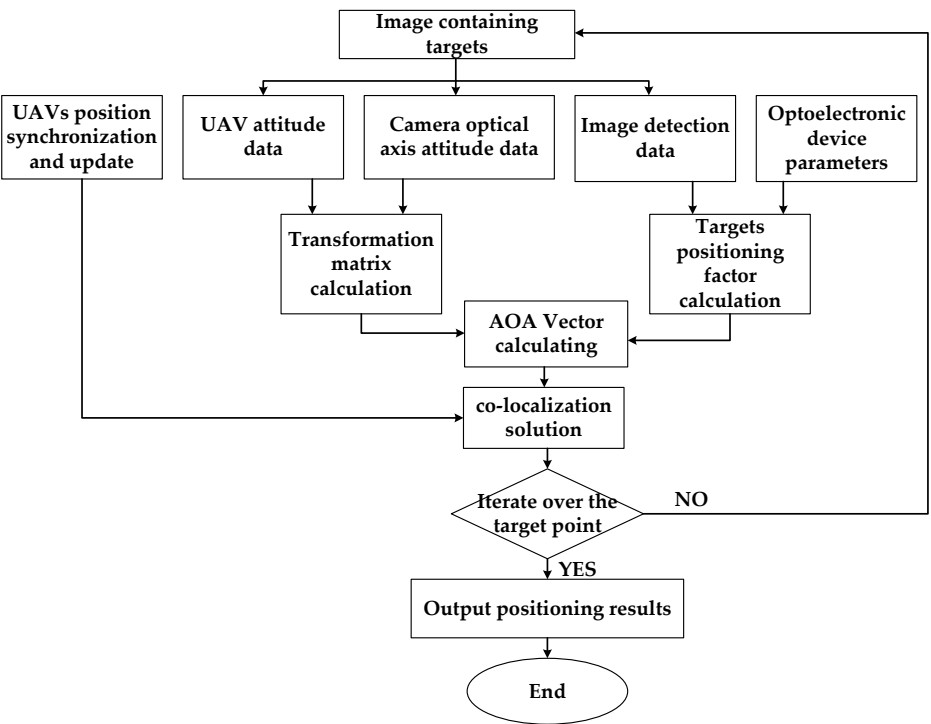

**Figure 2.** Flow chart of target co-location calculation.

### 3. Multi-UAV Target Co-Location Modeling

The set of UAVs involved in cooperative positioning is denoted by $S_n = \{S_{nj} | j = 1, 2, \ldots, N\}$, where N is the total number of UAVs involved in localization; the set of targets that may be scouted and located is denoted by $U_T = \{U_{T_i} | i = 1, 2, \ldots, K\}$, where K denotes the total number of targets that can be scouted and located.

#### 3.1. WGS-84 Earth Ellipsoid Model

The Earth ellipsoid is a mathematically defined Earth surface that approximates the geodetic level and serves as the reference framework for geodesy and global positioning techniques [9]. This reference also displays the WGS-84 Earth ellipsoid model's major parameters.

#### 3.2. Synchronization and Updating of Observational Position

The coordinates of the UAV coordinate system are derived from the image and the cooperative positioning of the target by the UAV, but because the positions of numerous UAVs are continually changing, the positions of multiple UAVs must be synchronized and updated.

It is expected that each UAV may collect its own geodetic coordinates and share their position with one another. Localization UAVs are classified into two types: leaders and followers. The mission planning technique assures that there is always one leader in the system to participate in positioning while the rest of the UAVs are followers. The method described in [10] is used in this paper to pick the leader aircraft.

This work develops a dynamic navigation coordinate system to ease the calculation. The dynamic navigation coordinate system ($O_n - X_n Y_n Z_n$) is defined as follows: the coordinate system's origin is solidly connected to the camera center of the lead aircraft, the $X_n$ axis is positively pointing to the north, the $Y_n$ axis is in the plumb plane and positively pointing to the sky, and the $Z_n$ axis follows the right-hand rule. The positions of the other UAVs in the dynamic navigation coordinate system are dynamically updated as the leader's position changes. Figure 3 depicts the position of each UAV in the dynamic navigation coordinate system at a given time.

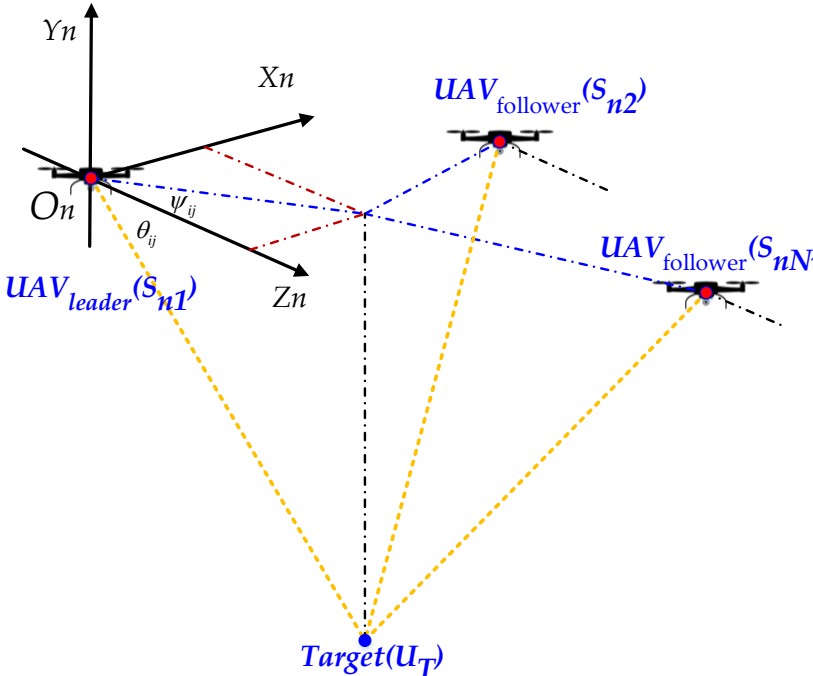

**Figure 3.** Schematic diagram of position synchronization and update based on dynamic navigation coordinate system.

The longitude, latitude, and altitude information for the UAV in the WGS-84 Earth ellipsoidal geodetic coordinate system is $\tau^g_{c(j)} = (\lambda_j, \varphi_j, h_j), (j = 1, 2, \ldots, N)$, and its position in the dynamic navigation coordinate system is $S_{nj}, (j = 1, 2, \ldots, N)$, where the lead aircraft coordinates are denoted as $\tau^g_{c(1)} = (\lambda_{0(1)}, \varphi_{0(1)}, h_{0(1)})$ and $S_{n1}$, respectively, and the positions of other UAVs in the dynamic navigation coordinate system are shown in Equation (1):

$$S_{nj} = C^n_e \cdot C^e_g \cdot (\tau^g_{c(j)} - \tau^g_{c(1)}), (j = 2, \ldots, N) \tag{1}$$

$$C^e_g = \begin{bmatrix} -\sin \varphi_{0(1)} \cos \lambda_{0(1)} & -\sin \varphi_{0(1)} \sin \lambda_{0(1)} & \cos \varphi_{0(1)} \\ \cos \varphi_{0(1)} \cos \lambda_{0(1)} & \cos \varphi_{0(1)} \sin \lambda_{0(1)} & \sin \varphi_{0(1)} \\ -\sin \lambda_{0(1)} & \cos \lambda_{0(1)} & 0 \end{bmatrix} \tag{2}$$

where $C^n_e$ denotes the ratio transformation of the geographical Cartesian coordinate system to the navigation coordinate system.

### 3.3. Image Based Localization Factor Solution Method

The target positioning solution process has to specify the coordinate system, angle, and coordinate system conversion matrix for a single image. Six coordinate systems: carrier coordinate system, servo stabilization coordinate system, electric-optic load system, and pixel coordinate system, as well as parameters like aircraft attitude angle, electric-optic load installation angle, servo frame angle, and look-down angle [11], are involved in addition to the definition of the coordinate systems shown in Sections 3.1 and 3.2.

By identifying the targets and using the coaxial image plane, the electric-optic load may be used to determine the pixel coordinates of each target. Through image detection data and optoelectronic device characteristics, the target information on the image can be determined. Give the symbol $\Theta_{Tij}$ to this information and define it as a target positioning factor.

First, build the camera coordinate system as depicted in Figure 4 and use the UAV's position $S_{nj}$ as the coordinate origin. In the navigation coordinate system, the coordinates of $S_{nj}$ are $(X_{ns(j)}, Y_{ns(j)}, Z_{ns(j)})$, those of the target point $A(i)$, which corresponds to the

image point $a(i)$, are $(X_{nA(i)}, Y_{nA(i)}, Z_{nA(i)})$, and the inverse equation of the common line equation is as follows:

$$
\begin{cases}
X_{nA(i)} - X_{ns(j)} = (Y_{nA(i)} - Y_{ns(j)}) \dfrac{c_{11(j)}X_{sa(i)} + c_{12(j)}Y_{sa(i)} + c_{13(j)}Z_{sa(i)}}{c_{21(j)}X_{sa(i)} + c_{22(j)}Y_{sa(i)} + c_{23(j)}Z_{sa(i)}} \\[2mm]
Z_{nA(i)} - Z_{ns(j)} = (Y_{nA(i)} - Y_{ns(j)}) \dfrac{c_{31(j)}X_{sa(i)} + c_{32(j)}Y_{sa(i)} + c_{33(j)}Z_{sa(i)}}{c_{21(j)}X_{sa(i)} + c_{22(j)}Y_{sa(i)} + c_{23(j)}Z_{sa(i)}}
\end{cases}
\tag{3}
$$

where $C_{mn(j)}(m \in [1,3], n \in [1,3])$, the conversion matrix between the navigation coordinate system and the UAV camera coordinate system, corresponds to the coefficients of $C^n_{c(j)}$.

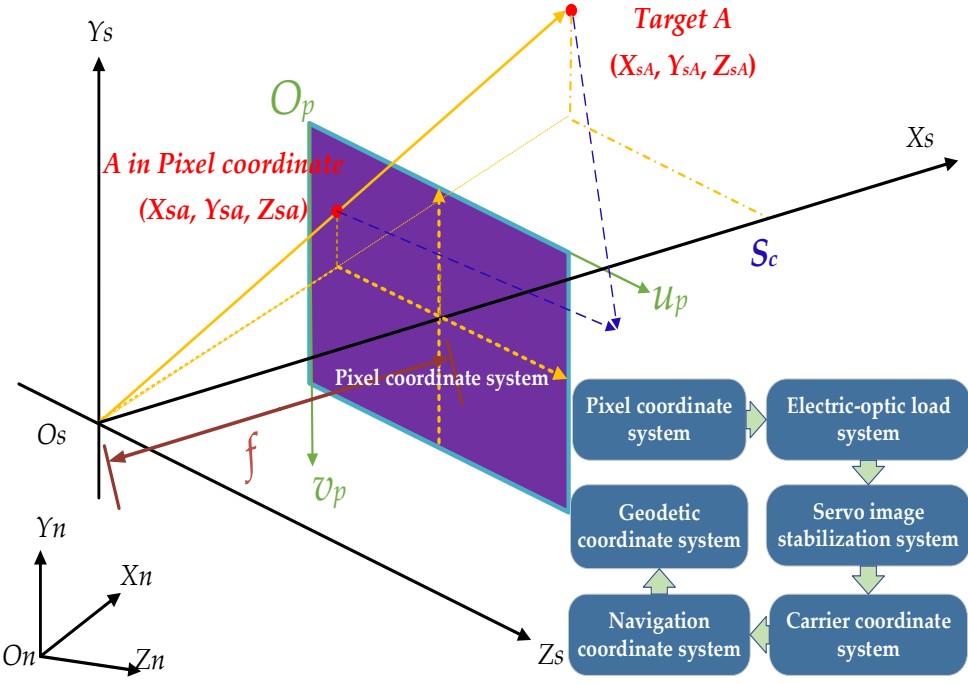

**Figure 4.** Schematic diagram of single-image based object detection.

Set the UAV electric-optic load's longitudinal and lateral resolution to $PxV_{max(j)} \times PxU_{max(j)}$, the half field of view angles in the longitudinal and lateral directions to $\alpha_{(j)1/2} \times \beta_{(j)1/2}$, the physical dimensions of the image elements in the longitudinal and lateral directions to $d_{v(j)} \times d_{u(j)}$, the focal length to $f_{(j)}$, and using the definition in Figure 4 we can obtain $f_{(i)} = X_{sa(i)}$, and the principal point of the image's position in the pixel coordinate system to $(u_{0(j)}, v_{0(j)})$. Target's pixel coordinates are $(u_i, v_i)$, $(i = 1, 2, \ldots, K)$, image point $a(i)$'s camera coordinates are $(X_{sa(i)}, Y_{sa(i)}, Z_{sa(i)})$, and the transformation between the two-dimensional image coordinate system and the camera coordinate system can be expressed as follows:

$$
\frac{1}{X_{sa(i)}}
\begin{pmatrix}
X_{sa(i)} \\
Y_{sa(i)} \\
Z_{sa(i)}
\end{pmatrix}
=
\begin{pmatrix}
1 \\
-(v_i - v_0)\dfrac{d_{v(j)}}{f_{(j)}} \\
(u_i - u_o)\dfrac{d_{u(j)}}{f_{(j)}}
\end{pmatrix}
\tag{4}
$$

The following formula can be derived in accordance with Figure 4 and its definition.

$$
\frac{d_{v(j)}}{f_{(j)}} = \frac{2\tan\left(\alpha_{(j)1/2}\right)}{PxV_{\max(j)}}
\tag{5}
$$

$$
\frac{d_{u(j)}}{f_{(j)}} = \frac{2\tan\left(\beta_{(j)1/2}\right)}{PxU_{\max(j)}}
\tag{6}
$$

The desired placement factor is the right-hand side of Equation (4), and the organized expression is as follows:

$$\Theta_{Tij} = \begin{bmatrix} 1 \\ -(v_i - v_{0(j)})\frac{2\tan(\alpha_{(j)1/2})}{PxV_{\max(j)}} \\ (u_i - u_{0(j)})\frac{2\tan(\beta_{(j)1/2})}{PxU_{\max(j)}} \end{bmatrix} \tag{7}$$

Define $C^n_{c(j)} = \begin{bmatrix} \omega^T_{nj} & \kappa^T_{nj} & \rho^T_{nj} \end{bmatrix}^T$, where $\omega_{nj} = \begin{pmatrix} C_{11(j)} & C_{12(j)} & C_{13(j)} \end{pmatrix}$, $\kappa_{nj} = \begin{pmatrix} C_{21(j)} & C_{22(j)} & C_{23(j)} \end{pmatrix}$, and $\rho_{nj} = \begin{pmatrix} C_{31(j)} & C_{32(j)} & C_{33(j)} \end{pmatrix}$. We can obtain the following by converting Equation (3):

$$\begin{cases} \frac{X_{nA(i)} - X_{ns(j)}}{Y_{nA(i)} - Y_{ns(j)}} = \frac{\omega_{nj} \times \Theta_{Tij}}{\kappa_{nj} \times \Theta_{Tij}} \\ \frac{Z_{nA(i)} - Z_{ns(j)}}{Y_{nA(i)} - Y_{ns(j)}} = \frac{\rho_{nj} \times \Theta_{Tij}}{\kappa_{nj} \times \Theta_{Tij}} \end{cases} \tag{8}$$

The internal orientation components and target pixel coordinates used in the UAV $S_{nj}$'s reconnaissance of the target $U_{T_i}$ are represented by $\Theta_{Tij}$ in Equation (7). It is clear that for a specific kind of electric-optic load, the internal orientation elements $\alpha_{(j)1/2}$, $\beta_{(j)1/2}$, $PxV_{max(j)}$, $PxU_{max(j)}$, $u_{0(j)}$ and $v_{0(j)}$ have constant values and that the localization factor only varies with the target pixel coordinates and is unaffected by the actual size and focal length of the image element.

### 3.4. Image-Based AOA Vector Solution Process

The AOA can be solved if the current position and attitude of the UAV and the orientation factor within the target are known as follows:

$V_{ij} = [\theta_{ij}(k), \psi_{ij}(k)]^T$ represents the AOA Vector. The following equation is obtained using Equation (8).

$$\tan V_{ij} = \begin{bmatrix} \sigma_{target(i)} \cdot \frac{1}{\|\zeta_{ij}\|_2} \\ \varepsilon_{ij} \end{bmatrix} \tag{9}$$

$$\zeta_{ij} = \begin{bmatrix} \frac{\omega_{nj} \times \Theta_{Tij}}{\kappa_{nj} \times \Theta_{Tij}} & \frac{\rho_{nj} \times \Theta_{Tij}}{\kappa_{nj} \times \Theta_{Tij}} \end{bmatrix}^T \tag{10}$$

$$\varepsilon_{ij} = \frac{\omega_{nj} \times \Theta_{Tij}}{\rho_{nj} \times \Theta_{Tij}} \tag{11}$$

From Equation (9) we obtain:

$$V_{ij} = \begin{bmatrix} \sigma_{target(i)} \cdot \tan^{-1}\left(\frac{1}{\|\zeta_{ij}\|_2}\right) \\ \tan^{-1}(\varepsilon_{ij}) \end{bmatrix} \tag{12}$$

Define $\sigma_{target(i)}$ as follows:

$$\sigma_{target(i)} = \begin{cases} 1, & \textit{Aerial Target} \\ -1, & \textit{Ground Target} \end{cases} \tag{13}$$

As evident from Equations (9)–(13), targets positioning factor $\Theta_{Tij}$, the transformation matrix $C^n_{c(j)}$ and the target coefficient $\sigma_{target(i)}$ have the biggest effects on the AOA vector.

This paper performs ground reconnaissance operations with $\sigma_{target(i)}$ set to $-1$.

The AOA Vector for N UAVs is:

$$V_{S_N} = [V_{i1}, \ldots, V_{ij} \ldots, V_{iN}]^T \tag{14}$$

*3.5. Co-Location Solution Model*

The targets' location coordinates are $U_{T_i} = \begin{bmatrix} x_{T_i}, y_{T_i}, z_{T_i} \end{bmatrix}^T (i = 1, 2, \ldots, K)$. Figure 3 depicts the relationship between the UAV $S_{nj}$ and the target $U_{T_i}$, with $R_{ij}$ serving as a measure of their separation.

According to Figure 3 and Equations (8) and (12), the following equation can be obtained:

$$U_{Ti} - S_{nj} = R_{ij} \cdot \begin{bmatrix} \cos \theta_{ij} \cdot \sin \psi_{ij} \\ \sin \theta_{ij} \\ \cos \theta_{ij} \cdot \cos \psi_{ij} \end{bmatrix} \tag{15}$$

Define $\tau_{\theta_{ij},\psi_{ij}} = \begin{bmatrix} \tau_1^{\theta_{ij},\psi_{ij}} & \tau_2^{\theta_{ij}} & \tau_3^{\theta_{ij},\psi_{ij}} \end{bmatrix}^T$, then:

$$\begin{bmatrix} \tau_1^{\theta_{ij},\psi_{ij}} \\ \tau_2^{\theta_{ij}} \\ \tau_3^{\theta_{ij},\psi_{ij}} \end{bmatrix} = \begin{bmatrix} \cos \theta_{ij} \cdot \sin \psi_{ij} \\ \sin \theta_{ij} \\ \cos \theta_{ij} \cdot \cos \psi_{ij} \end{bmatrix} \tag{16}$$

The procedure suggested in [12] allows us to roughly eliminate $R_{ij}$:

$$\Phi_{ij} \times U_{Ti} = \Phi_{ij} \times S_{nj}, (i = 1, 2, \ldots, K, j = 1, 2, \ldots, N) \tag{17}$$

$$\Phi_{ij} = \begin{pmatrix} \left(\tau_1^{\theta_{ij},\psi_{ij}}\right)^2 - 1 & \tau_1^{\theta_{ij},\psi_{ij}} \tau_2^{\theta_{ij}} & \tau_1^{\theta_{ij},\psi_{ij}} \tau_3^{\theta_{ij},\psi_{ij}} \\ \cdots & \left(\tau_2^{\theta_{ij}}\right)^2 - 1 & \tau_2^{\theta_{ij}} \tau_3^{\theta_{ij},\psi_{ij}} \\ \cdots & \cdots & \left(\tau_3^{\theta_{ij},\psi_{ij}}\right)^2 - 1 \end{pmatrix} \tag{18}$$

where $\Phi_{ij}$ is the symmetric matrix and $rank(\Phi_{ij}) = 2$, and thus Equation (17) requires more equations than necessary to meet the target position's solution.

For UAVs involved in localization $S_n$, there exists:

$$\begin{bmatrix} \Phi_{i1} \\ \Phi_{i2} \\ \cdots \\ \Phi_{iN} \end{bmatrix} \times U_{Ti} = \begin{bmatrix} \Phi_{i1} \times S_{n1} \\ \Phi_{i2} \times S_{n2} \\ \cdots \\ \Phi_{iN} \times S_{nN} \end{bmatrix}, N \geq 2 \tag{19}$$

It is possible to determine the target location coordinates by using Equation (1), Equation (7), Equation (12) and Equation (19). By converting the Earth's Cartesian coordinate system to the geodetic coordinate system and iteratively calculating the final geodetic coordinates, satisfying the accuracy longitude, latitude, and altitude as $(\lambda_{Ti}, \phi_{Ti}, h_{Ti})$, and the result $U_{T_i}$ is the coordinates of the navigation system of each target calculated with the coordinates of the UAV $S_{nj}$ in the navigation coordinate system as the reference point. After converting the measure to degrees, the targets' Earth coordinates are finally discovered.

## 4. Collaborative Positioning Error Model

*4.1. AOA Error Model Based on Image*

The look-down angle of the electric-optic load is $\theta_{xsj}$, with error $\delta\theta_{xsj}(k)$, the installation angle of the electric-optic load is $[\phi_{pj}(k), \vartheta_{pj}(k), \gamma_{pj}(k)]$, with error $[\delta\phi_{pj}(k), \delta\vartheta_{pj}(k), \delta\gamma_{pj}(k)]$, the yaw, pitch, and roll angles at the camera moment are $[\phi_{bj}(k), \vartheta_{bj}(k), \gamma_{bj}(k)]$, with measurement errors $[\delta\phi_{bj}(k), \delta\vartheta_{bj}(k), \delta\gamma_{bj}(k)]$, the altitude and azimuth angles of the frame are $[\theta_{cj}(k), \psi_{cj}(k)]$, with error $[\delta\theta_{cj}(k), \delta\psi_{cj}(k)]$, and let the true value of the pixel coordinates of the target observed by UAV $S_{nj}$ at time k. It should be noted that since the navigation device and the electric-optic load are solidly coupled to reduce error, the installation angle is 0.

Then the observation vector is:

$$\hat{L}_k = \left[\hat{\theta}_{bj}(k), \hat{\phi}_{bj}(k), \hat{\gamma}_{bj}(k), \hat{\theta}_{cj}(k), \hat{\phi}_{cj}(k), \hat{\theta}_{xsj}(k)\right]^T \tag{20}$$

The observation measurement error is:

$$\delta L_k = \left[\delta\theta_{bj}(k), \delta\phi_{bj}(k), \delta\gamma_{bj}(k), \delta\theta_{cj}(k), \delta\phi_{cj}(k), \delta\theta_{xsj}(k)\right]^T \tag{21}$$

At time $k$, The measured value of AOA Vector $V_{S_N}$ is $\hat{V}_k$, The true value of AOA Vector $V_{S_N}$ is $V_k$, then we get:

$$\hat{V}_k = \left[\hat{V}_{i1}(k), \ldots, \hat{V}_{ij}(k) \ldots, \hat{V}_{iN}(k)\right]^T \tag{22}$$

$$V_k = \hat{V}_k + J_k \times \delta L_k + \sigma_k^J \tag{23}$$

where $J_k$ is the Jacobian matrix, $\sigma_k^J$ the residual vector, and each value in $\sigma_k^J$ is the residual of a single observation from its standardized value.

The calculation of the matrix $J_k$ is performed below.

To facilitate the calculation, set the auxiliary variable as $\mathbf{M_{aux}} = \left[\overline{X}_j, \overline{Y}_j, \overline{Z}_j\right]^T$, $\Theta_{Tij}$ is normalized to obtain unit vector $\widetilde{\tau}_{c(i)}$, and $\widetilde{\tau}_{c(i)}$ is transformed by $C_{c(j)}^n$:

$$\mathbf{M_{aux}} = \begin{bmatrix} \overline{X}_j \\ \overline{Y}_j \\ \overline{Z}_j \end{bmatrix} = C_{c(j)}^n \cdot \widetilde{\tau}_{c(i)} (i = 1, 2, \ldots, M), (j = 1, 2, \ldots, N) \tag{24}$$

Equation (24) is transformed to obtain a new expression for the AOA Vector.

$$V_k = \begin{bmatrix} \cdots & \arcsin\overline{Y}_j & \arctan\dfrac{\overline{X}_j}{\overline{Z}_j} & \cdots & \arcsin\overline{Y}_N & \arctan\dfrac{\overline{X}_N}{\overline{Z}_N} \end{bmatrix}^T \tag{25}$$

A linearized transformation of Equation (23) yields $J_k$:

$$J_k = \begin{bmatrix} f_1(\theta_{b1}) & f_1(\phi_{b1}) & f_1(\gamma_{b1}) & f_1(\theta_{c1}) & f_1(\phi_{c1}) & f_1(\theta_{xs1}) \\ g_1(\theta_{b1}) & g_1(\phi_{b1}) & g_1(\gamma_{b1}) & g_1(\theta_{c1}) & g_1(\phi_{c1}) & g_1(\theta_{xs1}) \\ \cdots & \cdots & \cdots & \cdots & \cdots & \cdots \\ f_j(\theta_{bj}) & f_j(\phi_{bj}) & f_j(\gamma_{bj}) & f_j(\theta_{cj}) & f_j(\phi_{cj}) & f_j(\theta_{xsj}) \\ g_j(\theta_{bj}) & g_j(\phi_{bj}) & g_j(\gamma_{bj}) & g_j(\theta_{cj}) & g_j(\phi_{cj}) & g_j(\theta_{xsj}) \\ \cdots & \cdots & \cdots & \cdots & \cdots & \cdots \end{bmatrix}$$

$$define:$$

$$f_j(x) = K_j\frac{\partial\overline{Y}_j}{\partial v_j}, v_j = \theta_{bj}, \phi_{bj}, \gamma_{bj}, \theta_{pj}, \phi_{pj}, \gamma_{pj}, \theta_{cj}, \phi_{cj}, \theta_{xsj}$$

$$g_j(x) = W_j\left(\frac{\partial\overline{X}_j}{\partial v_j}\overline{Z}_j - \frac{\partial\overline{Z}_j}{\partial v_j}\overline{X}_j\right), v_j = \theta_{bj}, \phi_{bj}, \gamma_{bj}, \theta_{pj}, \phi_{pj}, \gamma_{pj}, \theta_{cj}, \phi_{cj}, \theta_{xsj}$$

$$\tag{26}$$

where: $K_j = \dfrac{1}{\sqrt{1-\overline{Y}_j^2}}$, $W_j = \dfrac{1}{\overline{X}_j^2+\overline{Z}_j^2}$

$$\frac{\partial\mathbf{M_{aux}}}{\partial v_j} = \frac{\partial C_{c(j)}^n}{\partial v_j}\widetilde{\tau}_{c(i)}, v_j = \theta_{bj}, \phi_{bj}, \gamma_{bj}, \theta_{pj}, \phi_{pj}, \gamma_{pj}, \theta_{cj}, \phi_{cj}, \theta_{xsj} \tag{27}$$

### 4.2. Collaborative Positioning Error Model Based on PDOP

Position Dilution of Precision, or PDOP, is simply a measure of how accurate a location is, and in a satellite positioning system, the degree of the PDOP value indicates how well-distributed the satellite terminals are [13]. During the cooperative positioning of several

UAVs, the DOP value of each measurement site in the reconnaissance region is influenced, with reduced DOP values frequently resulting in higher positioning accuracy [14]. In order to analyze the error range of cooperative positioning and to suggest optimization ideas, PDOP simulation and analysis of the cooperative positioning of numerous UAVs are employed in this study. The derivation formula in question is displayed below.

Assuming that the target is indeed in the position $[x_{Ti}, y_{Ti}, z_{Ti}]^T, (i = 2, 3, \ldots, M)$, the target solution is in the position $[\hat{x}_{Ti}, \hat{y}_{Ti}, \hat{z}_{Ti}]^T, (i = 2, 3, \ldots, M)$, and we obtain:

$$\delta X_k = \begin{bmatrix} x_{Ti} \\ y_{Ti} \\ z_{Ti} \end{bmatrix} - \begin{bmatrix} \hat{x}_{Ti} \\ \hat{y}_{Ti} \\ \hat{z}_{Ti} \end{bmatrix} \tag{28}$$

$$V_k = \hat{V}_k + H_k \times \delta X_k + \sigma_k^H \tag{29}$$

$H_k$ is the Jacobian matrix, and $\sigma_k^H$ is a vector of residuals between the observed values and the standard values. Each value in this vector represents the residual of a single observation compared to its standard value.

According to Equation (29), $H_k$ is obtained as:

$$H_k = \begin{bmatrix} -\frac{(x_{Ti}-x_{n1})(y_{Ti}-y_{n1})}{L_1 r_1^2} & \frac{L_1}{r_1^2} & -\frac{(z_{Ti}-z_{n1})(y_{Ti}-y_{n1})}{L_1 r_1^2} \\ \frac{z_{Ti}-z_{n1}}{L_1^2} & 0 & -\frac{(x_{Ti}-x_{n1})}{L_1^2} \cdot sign(z_{Ti}-z_{n1}) \\ \cdots & \cdots & \cdots \\ -\frac{(x_{Ti}-x_{nj})(y_{Ti}-y_{nj})}{L_j r_j^2} & \frac{L_j}{r_j^2} & -\frac{(z_{Ti}-z_{nj})(y_{Ti}-y_{nj})}{L_j r_j^2} \\ \frac{z_{Ti}-z_{nj}}{L_j^2} & 0 & -\frac{(x_{Ti}-x_{nj})}{L_j^2} \cdot sign(z_{Ti}-z_{nj}) \\ \cdots & \cdots & \cdots \end{bmatrix} \tag{30}$$

The covariance array of the error $\delta X_k$ is:

$$G_{\delta X} = E[\delta X_k \delta X_k^T] = B_{\delta X} \left\{ E[\delta V_k \delta V_k^T] \right\} B_{\delta X}^T \tag{31}$$

$$B_{\delta X} = \left(H_k^T H_k\right)^{-1} H_k^T \tag{32}$$

$$PDOP = \sqrt{trac(G_{\delta X})} \tag{33}$$

The PDOP solution procedure for cooperative localization of multiple UAV targets is provided by Equations (30)–(33).

### 4.3. Error Analysis

There are two sections to the error analysis. One is the external orientation element, or the UAV's flying status at the time of target launch, as illustrated in Table 1. The other is the internal orientation element, which is represented by the target pixel coordinates at the time of target launch in Tables 2 and 3, which displays the distribution of the measurement errors for the external orientation element.

The simulation is run under the conditions indicated in Table 1 to ensure that the error model is accurate, and the additional parameters are shown in Tables 2 and 3.

The location factor can be calculated as $\Theta_{Tij} \approx \begin{bmatrix} 1 & 0 & 0 \end{bmatrix}^T$ from Equation (7), assuming that the target is close to the center of the image. According to the circumstances outlined in [11], the error when the target is at the edge of the image is greater than the error when the target is in the center of the image, so four places in Figure 5A–D were chosen for a Monte Carlo simulation using arithmetic. The look-down angle of view was set at $-90°$, and the relative altitude of the flight was 3000 m.

**Table 1.** Observation parameters.

| Parameter | Signal | Numerical Value |
|---|---|---|
| pitch/° | $\vartheta_{bj}$ | 1.2 |
| yaw/° | $\phi_{bj}$ | 0 |
| roll/° | $\gamma_{bj}$ | −0.5 |
| Installation pitch/° | $\vartheta_{pj}$ | 0 |
| Installation yaw/° | $\phi_{pj}$ | 0 |
| Installation roll/° | $\gamma_{pj}$ | 0 |
| Frame altitude angle/° | $\vartheta_{cj}$ | −1.2 |
| Frame azimuth angle/° | $\phi_{cj}$ | 0.5 |
| Look down angle/° | $\vartheta_{xsj}$ | −90 |

**Table 2.** Electro-optical reconnaissance equipment parameters.

| Parameter | Signal | Value |
|---|---|---|
| Field of view | $\beta_{(j)} \times \alpha_{(j)}$ | $28° \times 21°$ |
| Resolution | $PxU_{max(j)} \times PxV_{max(j)}$ | $4096 \times 3072$ |
| Horizontal half field of view | $\beta_{(j)1/2}$ | 14° |
| Vertical half field of view | $\alpha_{(j)1/2}$ | 10.5° |
| Principal point position | $(u_{0(j)}, v_{0(j)})$ | (2047, 1535) |

**Table 3.** Observation parameters.

| Parameter | Signal | Error Range |
|---|---|---|
| Pitch error/° | $\delta\vartheta_{bj}$ | N (0, 0.8) |
| Yaw error/° | $\delta\phi_{bj}$ | N (0, 0.8) |
| Roll error/° | $\delta\gamma_{bj}$ | N (0, 0.8) |
| Installation pitch error/° | $\delta\vartheta_{pj}$ | N (0, 0.2) |
| Installation yaw error/° | $\delta\phi_{pj}$ | N (0, 0.2) |
| Installation roll error/° | $\delta\gamma_{pj}$ | N (0, 0.2) |
| Frame altitude angle error/° | $\delta\vartheta_{cj}$ | N (0, 0.1) |
| Frame azimuth angle error/° | $\delta\phi_{cj}$ | N (0, 0.1) |

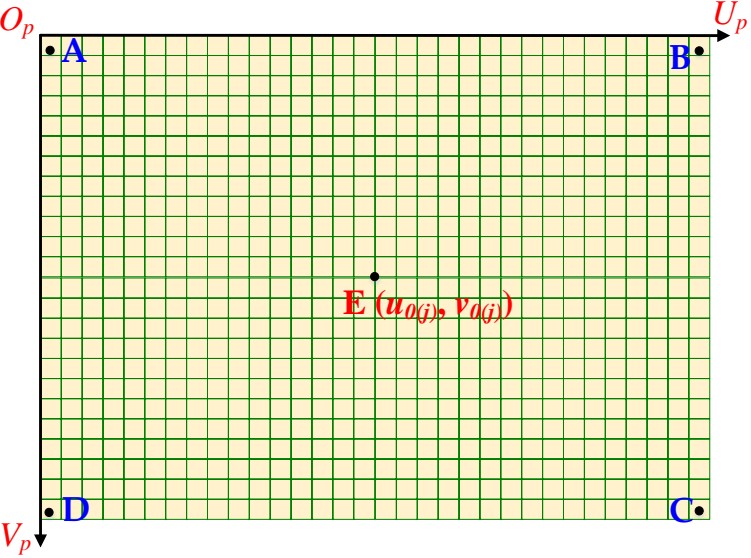

**Figure 5.** Points selected for Monte Carlo simulation to obtain maximum error. A, B, C and D correspond to the maximum error position and E corresponds to the minimum error position.

Figures 6–9 display the distribution of AOA errors for the target sites when the parameter values in Table 1 are used. Table 4 displays the computation results and observational parameters.

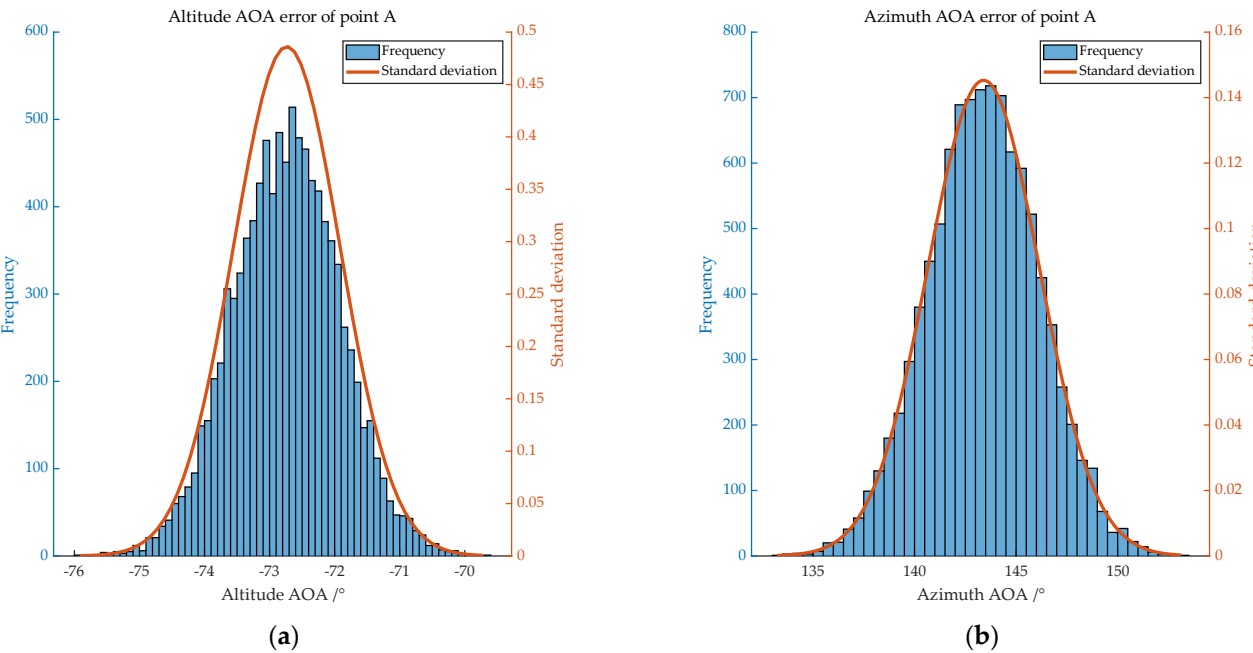

(**a**)  (**b**)

**Figure 6.** Distribution of AOA errors of A. (**a**) shows the distribution of altitude AOA error, with mean −72.747 and standard deviation 0.810; (**b**) shows the distribution of azimuth AOA error, with mean 143.377 and standard deviation 2.734.

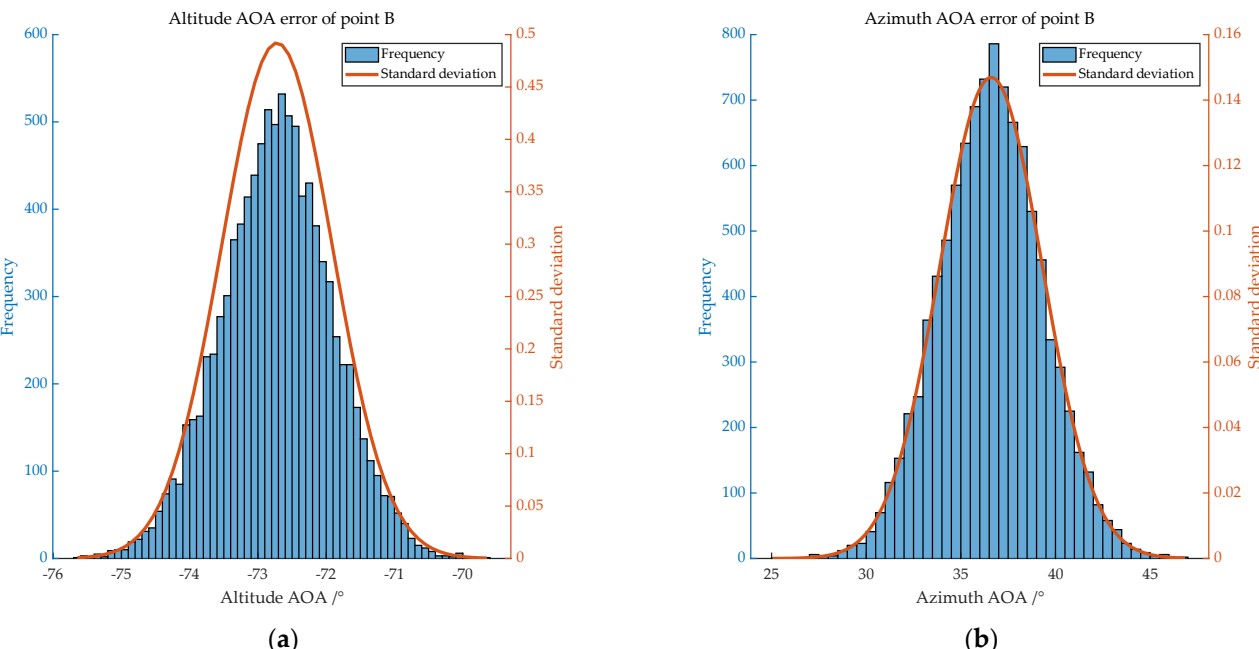

(**a**)  (**b**)

**Figure 7.** Distribution of AOA errors of B. (**a**) shows the distribution of altitude AOA error, with mean −72.733 and standard deviation 0.803; (**b**) shows the distribution of azimuth AOA error, with mean 36.590 and standard deviation 2.701.

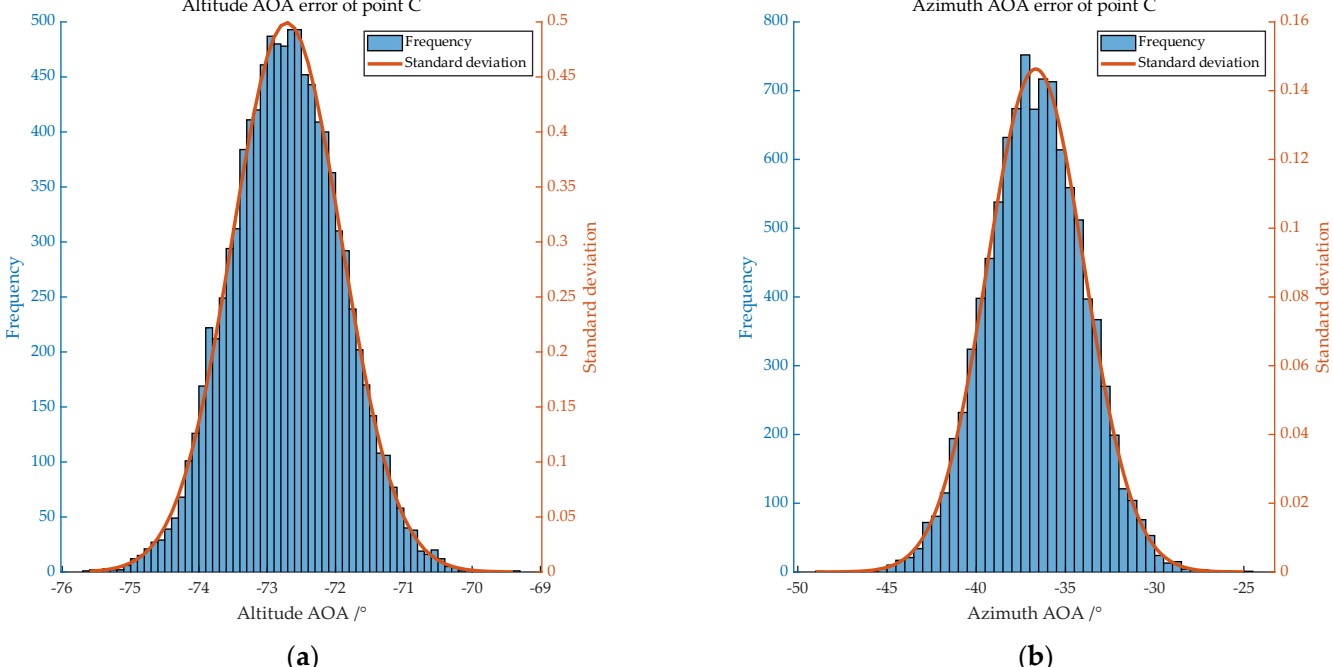

(**a**)  (**b**)

**Figure 8.** Distribution of AOA errors of C. (**a**) shows the distribution of altitude AOA error, with mean −72.702 and standard deviation 0.799; (**b**) shows the distribution of azimuth AOA error, with mean −36.655 and standard deviation 2.715.

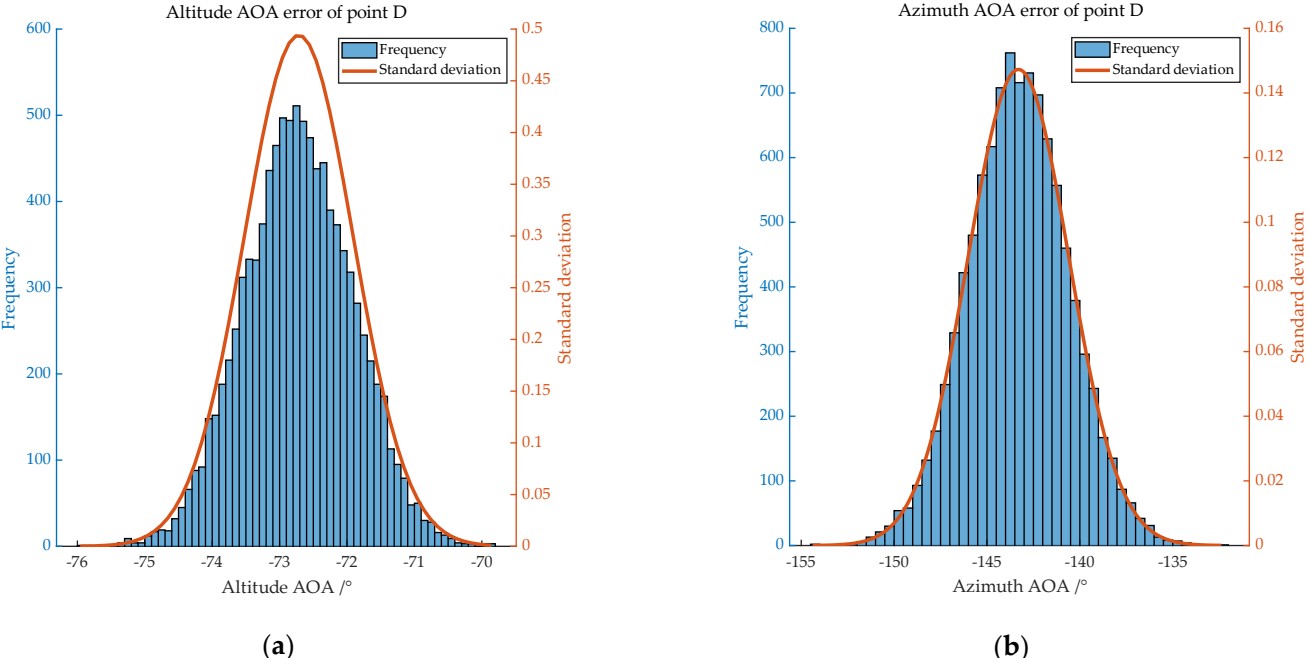

(**a**)  (**b**)

**Figure 9.** Distribution of AOA errors of D. (**a**) shows the distribution of altitude AOA error, with mean −72.726 and standard deviation 0.808; (**b**) shows the distribution of azimuth AOA error, with mean −143.310 and standard deviation 2.727.

**Table 4.** AOA of measurements at each point under condition 1.

| ID | Point | Pixel Coordinate | Distribution of AOA Altitude Angle/° (Monte Carlo) | Distribution of AOA Altitude Angle/° (PDOP) | Distribution of AOA Azimuth Angle/° (Monte Carlo) | Distribution of AOA Altitude Angle/° (PDOP) |
|----|-------|------------------|-----------------------------------------------------|----------------------------------------------|----------------------------------------------------|----------------------------------------------|
| 1 | A | (0, 0) | N (−72.747, 0.810) | N (−72.75, 0.806) | N (143.377, 2.734) | N (143.38, 2.717) |
| 2 | B | (4096, 0) | N (−72.733, 0.803) | N (−72.74, 0.806) | N (36.590, 2.701) | N (36.59, 2.715) |
| 3 | C | (4096, 3072) | N (−72.702, 0.799) | N (−72.73, 0.806) | N (−36.655, 2.715) | N (−36.63, 2.714) |
| 4 | D | (0, 3072) | N (−72.726, 0.808) | N (−72.74, 0.806) | N (−143.310, 2.727) | N (−143.34, 2.716) |

According to the experiments, under ideal flight conditions, the target's image-based AOA azimuth angle and altitude angles should both not exceed 2.73° and 0.81°, respectively.

Based on the calculation method of PDOP value proposed in Sections 4.1 and 4.2, the multi-aircraft cooperative localization error analysis is carried out. Firstly, two UAVs are set up for target localization; the position coordinates of the UAVs are shown in Table 5, the position error is 0 m, and the flight altitude is 3000 m. According to the parameters in Table 2, it can be known that the size of the corresponding area of the captured image is calculated to be about 1400 m × 1200 m when flying at a relative altitude of 3000 m. In this area, the PDOP value under any position can be calculated and plotted into a contour distribution map, as shown in Figure 10.

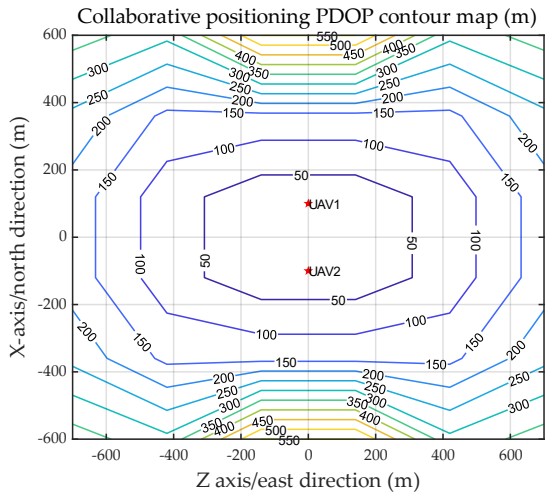

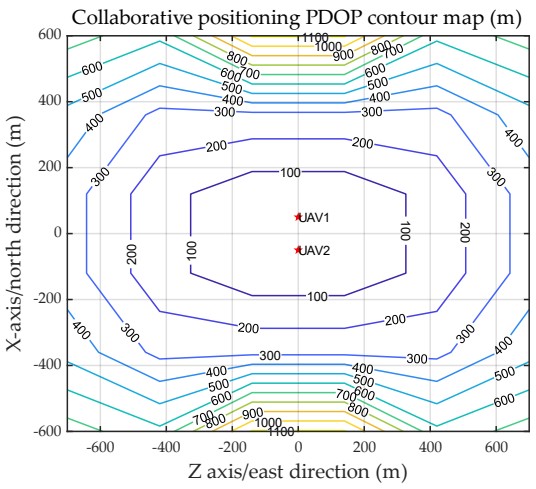

**Figure 10.** Two-UAV cooperative reconnaissance PDOP simulation (Baseline: 200 m and 100 m).

**Table 5.** Coordinate distribution of UAVs.

| Point Coordinates | Symbol | Value 1/m | Value 2/m |
|---|---|---|---|
| UAV1 | (x1, y1, z1) | (50, 0, 0) | (25, 0, 0) |
| UAV2 | (x2, y2, z2) | (−50, 0, 0) | (−25, 0, 0) |

The results of cooperative localization of two UAVs show that when the distance between UAVs is 200 m and 100 m, the minimum value of PDOP distribution is 50 m and 100 m, respectively, i.e., the smaller the spacing is, the larger the localization error is. In addition, the target localization accuracy is also related to the distribution of UAVs; therefore, four UAVs are set up for target localization, and their errors are analyzed. Assuming that four UAVs fly at a relative height of 3000 m, the position coordinates in the 2D plane are shown in Table 6. The calculation results are shown in Figures 11 and 12.

**Table 6.** Coordinate distribution of UAVs.

| Point Coordinates | Symbol | Square Formation Flying/m | Diamond Formation Flying/m |
|---|---|---|---|
| UAV1 | (x1, z1) | (100, 100) | (0, 100) |
| UAV2 | (x2, z2) | (100, −100) | (0, −100) |
| UAV3 | (x3, z3) | (−100, 100) | (200, 0) |
| UAV4 | (x4, z4) | (−100, −100) | (−200, 0) |

PDOP Contour line is distributed as follows.

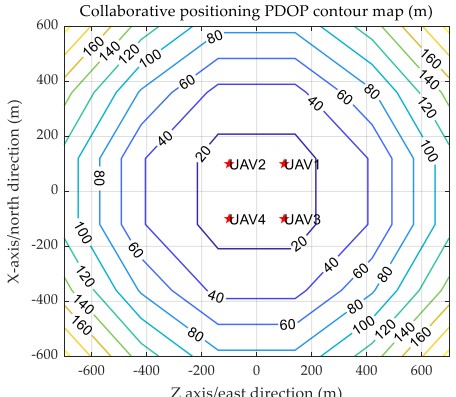

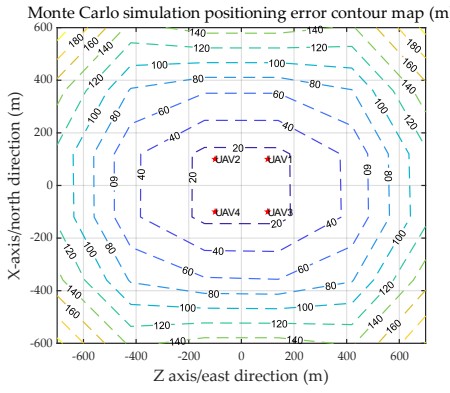

**Figure 11.** PDOP and Monte Carlo simulation position error conture map of square formation flying.

Figures 11 and 12 compare the PDOP values of four unmanned aerial vehicles' target collaborative positioning under different flying modes with Monte Carlo shooting simulation results. The errors between the two are roughly equal, with a minimum placement inaccuracy of about 20 m.

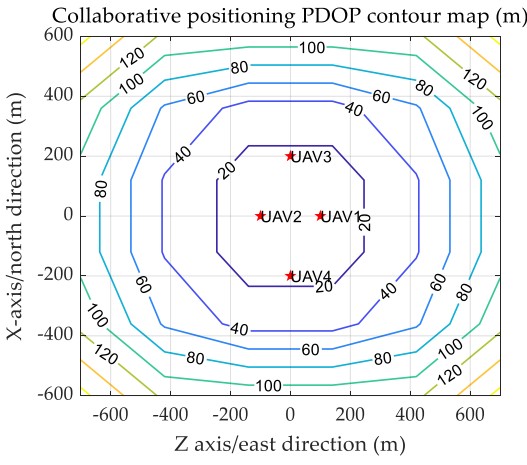

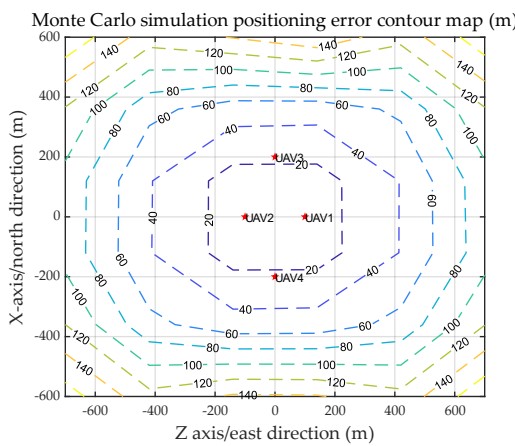

**Figure 12.** PDOP and Monte Carlo simulation position error conture map of diamond formation flying.

In summary, utilizing the collaborative positioning model of many unmanned aerial vehicles, a PDOP-based error computation model was constructed, and it was validated using Monte Carlo simulation. The simulation findings show that the baseline has a significant impact on the collaborative positioning error of the two UAVs, with a minimal positioning error of roughly 50 m. The collaborative positioning error of four unmanned aerial vehicles is less affected by the formation mode under the same conditions, with a minimal positioning error of roughly 20 m. Filtering methods can increase the accuracy of error models.

## 5. Target Localization Method Based on Cubature Kalman Filter

Nonlinear filtering is used because the observation functions of both the time difference and the measured AOA observations are nonlinear functions [15,16]. Because of its linearization procedure, the Extended Kalman Filter (EKF) can only achieve a greater filtering performance provided the linearization error of the system's state and observation equations is small [17,18]. The particle filter (PF) algorithm, which has recently been developed, is a good algorithm for solving the nonlinear estimation problem [19,20]. As a result, the improvements in localization error by the EKF and PF algorithms were compared.

Arasaratnam and Haykin et al. proposed the Cubature Kalman Filter (CKF) algorithm [21] to solve the integration problem of nonlinear functions in filtering algorithms, which is similar to the Unscented Kalman Filter (UKF), first calculates the sampling points (called volume points), then calculates the one-step prediction of the volume points by the state equation, and then corrects the predicted value of the state by the quantitative update and the Kalman gain calculation. In comparison to the Unscented Kalman filter algorithm, the Cubature Kalman Filter algorithm obtains the volume points by calculating the spheri-

cal radial volume criterion without linearizing the state equation and directly transferring the volume points by the nonlinear state equation while ensuring that the weights are always positive. This improves the algorithm's robustness and accuracy [22–24].

According to the third-order spherical radial criterion, the number of volume points for an n-dimensional state vector is $m = 2n$, and the set of volume points is designated as:

$$\xi_j = \sqrt{\frac{m}{2}}[1]_j, j = 1, 2, \ldots, 2n \tag{34}$$

where $[1]_j$ denotes the $j^{th}$ volume point, i.e., the $j^{th}$ column of $[1]$, and $[1]$ can be expressed as:

$$[1] = \left[ \begin{pmatrix} 1 \\ 0 \\ \vdots \\ 0 \end{pmatrix} \cdots \begin{pmatrix} 0 \\ 0 \\ \vdots \\ 1 \end{pmatrix} \begin{pmatrix} -1 \\ 0 \\ \vdots \\ 0 \end{pmatrix} \cdots \begin{pmatrix} 0 \\ 0 \\ \vdots \\ -1 \end{pmatrix} \right] \tag{35}$$

The weights of each volume point are equal, as written:

$$\omega_j = \frac{1}{m} \tag{36}$$

For the following target state equation and the measurement equation:

$$\begin{aligned} X_k &= f(X_{k-1}, u_{k-1}) + w_{k-1} \\ Z_k &= h(X_k, u_k) + v_k \end{aligned} \tag{37}$$

The Cubature Kalman Filtering algorithm and the specific process are given below:
Step 1: Calculation of volume points.

$$\begin{cases} P_{k-1,k-1} = S_{k-1}S_{k-1}^T \\ \widetilde{\chi}_{k-1}^j = \hat{X}_{k-1} + S_{k-1}\xi_j \end{cases} \tag{38}$$

where a Cholesky decomposition of $P_{k-1,k-1}$ gives $S_{k-1}$.
Step 2: One-step prediction of volume points.

$$\widetilde{\chi}_{k,k-1}^{*j} = f(\widetilde{\chi}_{k-1}^j) \tag{39}$$

Step 3: Compute one-step prediction and covariance matrix of state quantities.

$$\begin{cases} \hat{X}_{k,k-1} = \dfrac{\sum\limits_{j=1}^m \widetilde{\chi}_{k,k-1}^{*j}}{m} \\ P_{k,k-1} = \dfrac{1}{m} \sum\limits_{j=1}^m w_j \widetilde{\chi}_{k,k-1}^{*j} \left(\widetilde{\chi}_{k,k-1}^{*j}\right)^T - \hat{X}_{k,k-1}\hat{X}_{k,k-1}^T + Q_{k-1} \end{cases} \tag{40}$$

Step 4: Calculation of new volume points based on one-step predicted values.

$$\begin{cases} P_{k,k-1} = S_{k,k-1}S_{k,k-1}^T \\ \widetilde{\chi}_{k,k-1}^j = \hat{X}_{k,k-1} + S_{k,k-1}\xi_j \end{cases} \tag{41}$$

Step 5: Observation prediction for new volume points.

$$Z_{k,k-1}^j = h(\widetilde{\chi}_{k,k-1}^j) \tag{42}$$

Step 6: Calculate the mean and covariance of the target observations weighted by the observation predictions of the volume points.

$$\hat{Z}_{k,k-1} = \frac{\sum\limits_{j=1}^{m} Z_{k,k-1}^{j}}{m} \tag{43}$$

$$P_{xz} = \frac{1}{m}\sum_{j=1}^{m} \tilde{\chi}_{k,k-1}^{j}\left(Z_{k,k-1}^{j}\right)^{T} - \hat{X}_{k,k-1}\hat{Z}_{k,k-1}^{T} \tag{44}$$

$$P_{zz} = \frac{1}{m}\sum_{j=1}^{m} Z_{k,k-1}^{j}\left(Z_{k,k-1}^{j}\right)^{T} - \hat{Z}_{k,k-1}\hat{Z}_{k,k-1}^{T} + R_k \tag{45}$$

Step 7: Calculating Kalman gain.

$$K_k = P_{xz}P_{zz}^{-1} \tag{46}$$

Step 8: Calculate system state update and covariance update.

$$\begin{cases} \hat{X}_k = \hat{X}_{k,k-1} + K_k(Z_k - \hat{Z}_{k,k-1}) \\ P_k = P_{k,k-1} - K_k P_{zz} K_k^T \end{cases} \tag{47}$$

The flow of the Cubature Kalman Filtering algorithm is shown in Figure 13.

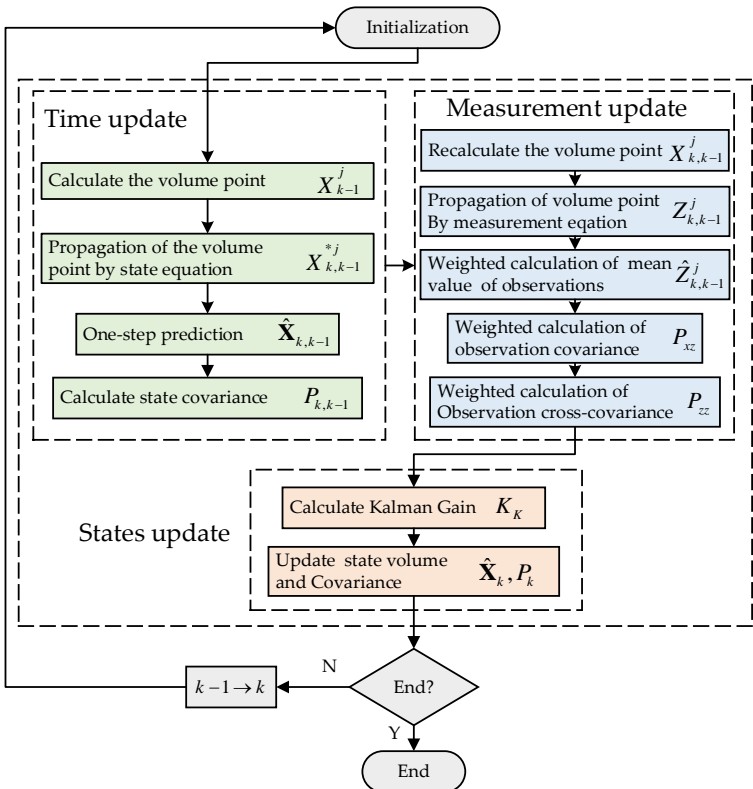

**Figure 13.** Flow of the Cubature Kalman Filtering algorithm.

## 6. Simulation and Analysis

### 6.1. Co-Localization Algorithm Verification

An external field test was performed to validate the co-localization algorithm's correctness. For the same ground identifier, twelve groups of UAV aerial photographs were selected under varied working conditions, with two images in each group, and the target

location was solved separately, yielding a total of six groups of target coordinate values. An example of a group of aerial images is shown in Figure 14.

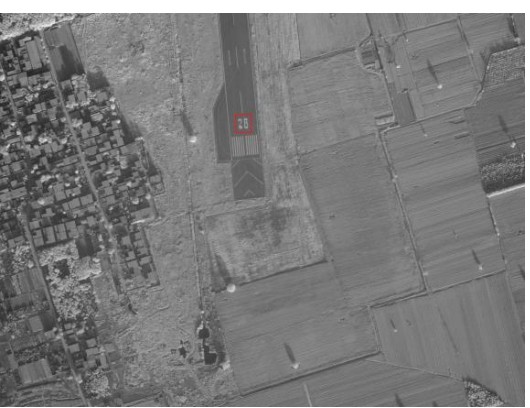

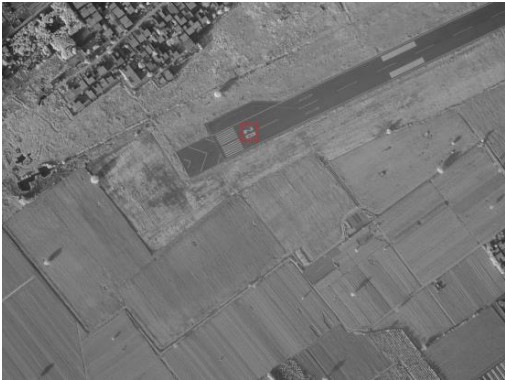

**Figure 14.** The ground images captured by two drones at the same time, with the number "28" in the red box as the ground identifier, and their true coordinates are known.

Figure 15 shows the target localization results (the status of UAV collaborative target localization is shown in Table 7, The calculation results of the target position in this state are shown in Table 8). The blue solid dots in the figure represent the true position of the ground identification (target), while the red solid dots represent the results of collaborative target localization by two machines. There are a total of six sets of data. It can be seen that the algorithm can accurately calculate the position of the target in the two-dimensional plane.

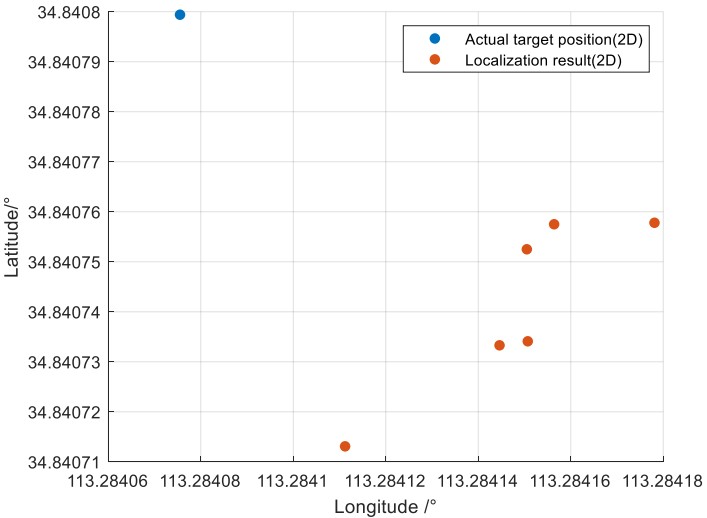

**Figure 15.** The distribution of actual target position and localization result in a two-dimensional plane.

**Table 7.** States of 12 UAVs using for localization.

| UAV ID | UAV Pose | | Installation Angle (Pitch, Yaw, Roll) | Frame Angle (Alt., Azim.) |
|---|---|---|---|---|
| | UAV Position (Lng., Lat., Alt.) | UAV Attitude (Pitch, Yaw, Roll) | | |
| 1 | 113.285672, 34.840742, 1988.7 | 3.25, 75.52, 1.32 | −90, 0, 0 | −3.27, −1.24 |
| 2 | 113.283782, 34.840863, 1987.8 | 3.41, 75.55, 2.98 | −90, 0, 0 | −3.52, −2.69 |
| 3 | 113.285258, 34.840542, 1986.6 | 2.75, 267.91, 0.89 | −90, 0, 0 | −2.76, −1.26 |
| 4 | 113.284669, 34.841472, 1980.9 | 3.36, 267.42, 1.62 | −90, 0, 0 | −3.37, −1.78 |
| 5 | 113.283662, 34.839757, 1978.0 | 2.49, 144.30, 0.40 | −90, 0, 0 | −2.54, −0.39 |
| 6 | 113.284839, 34.840663, 982.3 | 3.34, 144.85, −0.08 | −90, 0, 0 | −3.40, 0.10 |
| 7 | 113.283124, 34.840924, 982.2 | 3.75, 80.46, 3.67 | −90, 0, 0 | −3.79, −3.57 |
| 8 | 113.283091, 34.840818, 980.3 | 3.60, 79.11, 4.12 | −90, 0, 0 | −3.62, −4.13 |
| 9 | 113.285250, 34.840497, 980.1 | 6.15, 260.06, 3.47 | −90, 0, 0 | −6.23, −3.55 |
| 10 | 113.284723, 34.841950, 978.6 | 4.82, 261.98, −0.96 | −90, 0, 0 | −4.82, 0.81 |
| 11 | 113.284085, 34.840844, 976.0 | 2.79, 145.32, −1.25 | −90, 0, 0 | −2.79, 1.14 |
| 12 | 113.285672, 34.840742, 1988.7 | 2.81, 145.20, 2.55 | −90, 0, 0 | −2.87, −2.37 |

**Table 8.** Two UAVs collaborative target positioning results (first group ID: 1&5, second group ID: 2&6, third group ID: 3&4, fourth group ID: 7&9, fifth group ID: 8&11, sixth group ID: 10&12).

| Actual Target Position (2D) | | Localization Result (2D) | | Localization Error/m |
|---|---|---|---|---|
| Lng. | Lat. | Lng. | Lat. | |
| | | 113.284151 | 34.840752 | 8.6 |
| | | 113.284111 | 34.840713 | 10.1 |
| | | 113.284178 | 34.840758 | 10.5 |
| 113.284076 | 34.840799 | 113.284145 | 34.840733 | 9.8 |
| | | 113.284151 | 34.840734 | 9.9 |
| | | 113.284156 | 34.840757 | 8.7 |

*6.2. Multi-UAV Co-Location and Tracking*

Simulations for fixed and moving targets are discussed in this section, and these were used to validate the effectiveness of CKF. During the process of target discovery and localization by the four UAV observatories, the UAV moves along a specific trajectory and makes numerous observations of the target area. The observations from the first measurement are combined with the initial position where the UAV observatory begins positioning to obtain the initial position estimate of the target and the corresponding covariance matrix of the zero-mean estimation error as the initial estimate of the filtering algorithm, and the filtering algorithm is then used to process the multiple observations to obtain a more accurate estimate of the target.

Figure 3 shows the NUE (North-Up-East) coordinate system $O_n - X_n Y_n Z_n$, and Table 9 shows the beginning condition of the ground target, data related to the number of UAVs, initial status, and measurement errors.

**Table 9.** Co-location simulation parameters.

| Simulation Parameters | Range |
|---|---|
| Target initial state $x_T$ | Stationary: $[0, 0, 0, 0]^T$<br>Moving: $[0,\ 11.7851,\ 0,\ 11.7851]^T$ |
| Number | 4 |
| Error | 0.81° (AOA altitude angle)<br>2.73° (AOA azimuth angle) |
| Noise | Gaussian White Noise |
| Sampling step | 500 ms |

Ref. [25] provides the basic motion model of the target, and the efficiency of the Unscented Kalman Filter-based target localization approach is validated in this study using the target's constant linear and rotating motion.

The discrete constant velocity linear and rotating motion models of the target are shown in the following equations:

$$x_k^{cv} = \Phi_k^{cv} x_{k-1}^{cv} + W_k^{cv} \tag{48}$$

$$x_k^{ct} = \Phi_k^{ct} x_{k-1}^{ct} + W_k^{ct} \tag{49}$$

where $x_k^{cv} = \begin{bmatrix} x_k & \dot{x}_k \end{bmatrix}^T$ and $x_k^{ct} = \begin{bmatrix} \dot{x}_k & \dot{x}_k y_k & \dot{y}_k \end{bmatrix}^T$ are the state vectors of the target's constant linear and turning motion models, $\Phi_k^{cv}$ and $\Phi_k^{ct}$ the status transition matrices, and $W_k^{cv}$, $W_k^{ct}$ the system noise.

To model the positional adequacy and location error of the UAVs, a random normal error with a mean value of 0 and a standard deviation of 10 m is added to the position of the UAVs' path. Figure 16 depicts the error between the measuring position and the real position of the UAV.

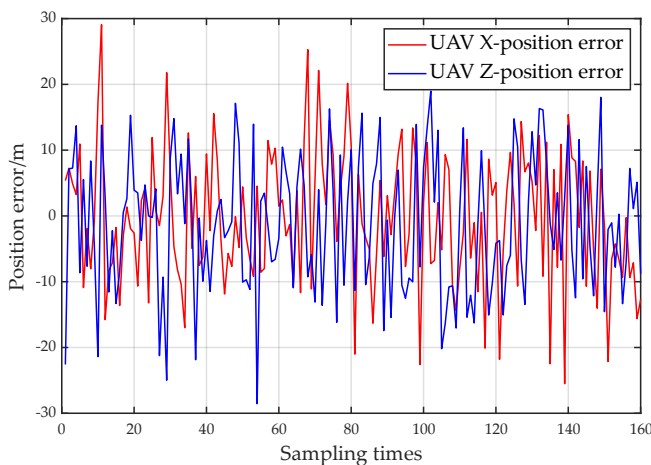

**Figure 16.** Diagrammatic representation of the noise interference on the drone's position, superimposed on the motion trajectory, with a mean of 0 and a standard deviation of 10 m.

Four UAVs approach the target from various angles. Figure 17 depicts the position result for a stationary object. The target position's positioning deviance quickly converges from tens of meters to less than ten meters.

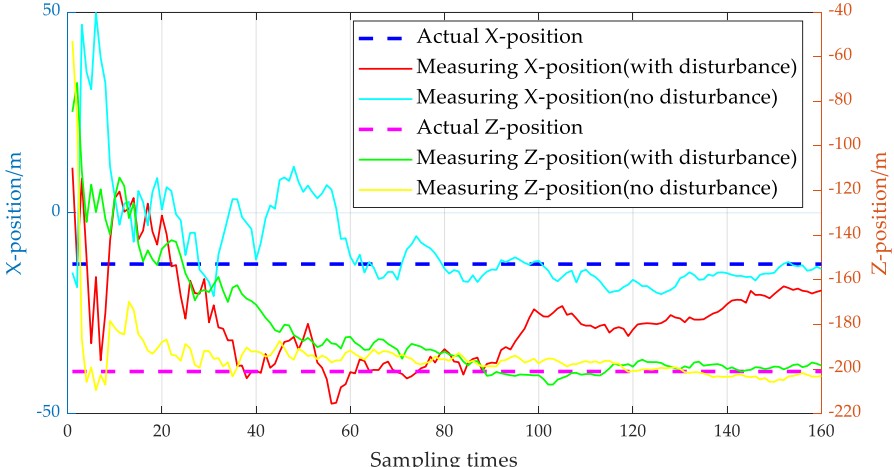

**Figure 17.** The position result for stationary target.

Assume that the target moves in a straight path and turns in a straight line and that one revolution of the motion takes 80 s. Four UAVs are programmed to proceed toward the target's initial position to finish the target's continual localization and tracking. Figure 18 depicts the target's and UAVs' respective motion trajectories and tracking at a certain time.

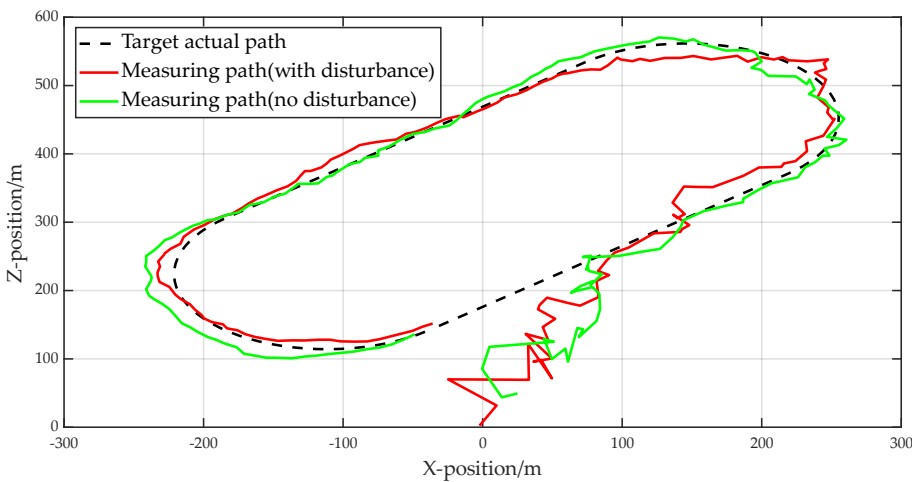

**Figure 18.** Moving target localization and tracking trajectory status (2D).

To account for the interference of random perturbations on positioning outcomes, 100 Monte Carlo simulations were run, and the RMSE (Root Mean Square Error) was employed as the accuracy judgment measure.

Figure 19 shows the measurement results of the position component when tracking a moving target, and it can be seen that the algorithm proposed in this paper can quickly converge and stably trac the target. Figure 20 shows the position and velocity RMS errors when tracking a moving target, and the tracking accuracy gradually improves with the change of time, the position error converges within 12 m, and the velocity error converges within 0.5 m/s, which also shows that the algorithm has high accuracy under external interference.

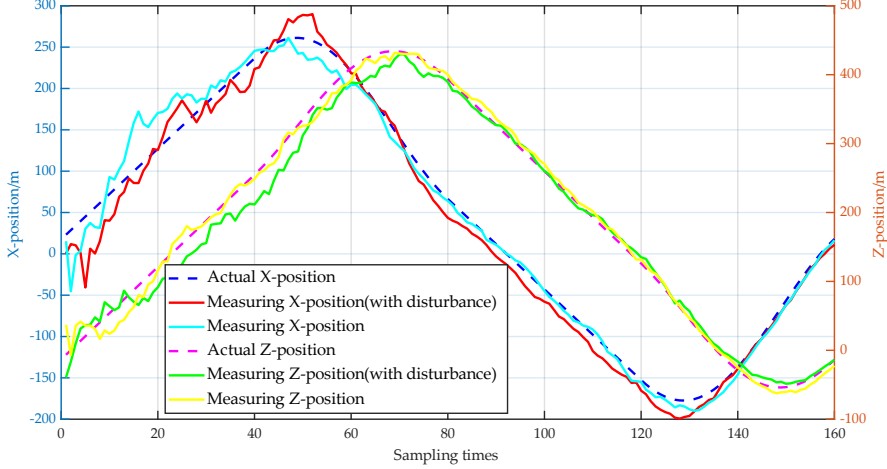

**Figure 19.** Tracking status of moving target position component (2D).

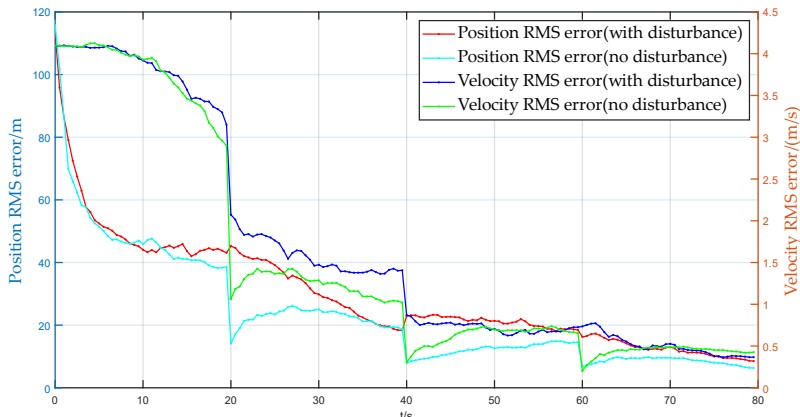

**Figure 20.** RMSE of target position and velocity prediction.

### 7. Conclusions

This paper's target co-localization approach is a localization method that does not rely on elevation and ranging information. It can calculate the positions of many targets at once, considerably improving UAV detecting capability. The approach is almost hardware-independent and thus appropriate for low-cost small UAV cluster systems. The error model is used in this study to examine the lowest target positioning error of 20 m at 3000 m relative flight altitude under typical flight conditions. This paper uses the traceless Kalman filtering algorithm to simulate and verify the stationary and moving targets, respectively, and the target localization accuracy is improved by 40% compared to the original one, and the target can be continuously tracked in the case of interference with a high degree of accuracy guaranteed.

**Author Contributions:** Conceptualization, M.D.; methodology, M.D.; software, M.D., T.W. and K.Z.; validation, M.D., H.Z. and T.W.; formal analysis, M.D. and K.Z.; investigation, T.W.; resources, H.Z.; data curation, H.Z.; writing—original draft preparation, M.D. and H.Z.; writing—review and editing, M.D. and H.Z.; visualization, M.D. and K.Z.; supervision, M.D.; project administration, M.D.; funding acquisition, M.D. All authors have read and agreed to the published version of the manuscript.

**Funding:** This research received no external funding.

**Data Availability Statement:** The data that support the findings of this study are available on request from the corresponding author, Minglei Du, upon reasonable request.

**Conflicts of Interest:** No potential conflict of interest was reported by the authors.

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
