# Peer review of "A Cooperative Target Localization Method Based on UAV Aerial Images"

_aerospace, doi:10.3390/aerospace10110943_

Round 1

Reviewer 1 Report

Comments and Suggestions for Authors

If I understand correctly, the point of the research is to design an algorithm for cooperative localisation of targets using image processing. Multiple UAVs are supposed to take pictures of the same target and exchange data to identify it.

I appreciate the table of symbols and the clear pictures.

"Drones are strongly connected to each other;" What do you mean? You should quantify the the link parameters that you need. How much data must be transferred from drone to drone for this to work? What are the latency requirements?

There is no mention of how does the algorithm cope with the drones oscillating around their position. It is impossible to keep a quad rotor UAV in a fixed position without fluctuations. Fluctuations are caused by wind, imbalance of the load or the rotors, rain... However, other kind of positional erros are kept into account. This is importante because it changes the performance of your stabilization mechanism and your Kalman filter.

"The Airplane mode of UAV is ideal Airplane mode" -> What does this mean?

When a new terminology is used, it should be clearly explained. 

Please consider sahring the code of your simulation after publication.

Figures 15 and 17 are not mentioned anywhere in the text.

The figure's captions are very short and do not explain what the figure is about.

There is no comparison between the target tracking/location algorithm presented in this paper and other solutions proposed in the literature.

I suggest to test your algorithm with actual pictures. You do not need drones for that. Just take pictures of the same object from multiple angles, even with a smartphone camera. You do not need to convert everything in WGS84 coordinates, a local/simpler coordinate system is enough to test this. This experiment can help understand better the constraints and limitations of your system, and to prove the soundness of your hypotesis.

At the end of the introduction, you should clearly state what the research questions are and how do y ou intend to answer them.

Comments on the Quality of English Language

"Under the conditions of passive positioning and no elevation information, how to achieve precise positioning of multiple targets through image information is a problem that small unmanned aerial vehicles need to solve." -> This can be rewritten as: "In passive positioning without elevation data, achieving precise positioning for multiple targets using image information is a challenge for small unmanned aerial vehicles."

"In addition, for small UAVs of several kilograms to tens of kilograms class, the increase in detection range or flight altitude makes the contradiction between the requirement to carry a radiosonde device and the overall design constraints of the UAV, as well as the contradiction between the miniaturization of the radiosonde device and the cost of its development, become more prominent." -> This sentence is convoluted and difficult to read. Can you please explain it in a shorter/easier way? What contradiction are you talking about? This is not clear from the text.

The whole paper needs accurate revision of the English. The text is extremely difficult to read and some part are impossible to understand. If you can, hire an editor and use Grammarly or similar.

Author Response

Dear reviewers,

Thank you so much for all your hard work on this article. In response to your feedback, we have made modifications one by one. The specific content is attached.

Reviewer 2 Report

Comments and Suggestions for Authors

The authors develop a new collaborative method to spatially localize targets with multiple cameras mounted on the members of a drone swarm. The article has added value and correctly explains and demonstrates the procedure. Although the results presented are only simulated.

In my opinion the article has merit to be published. However, the quality would be increased if some presentation aspects were improved. Below I put some suggestions:

1-      Explain better the new features of the proposal in the introduction.

2-      Since the article talks about location with UAVs in general, avoid using military nomenclature (“patrol bombs”, “small loitering munitions”…).

3-      Improve the bibliography. There are several reference references on basic technology that are in Chinese. It would be better to add one in English that also has these contents.

4-      Review the nomenclature and define all the terms in section 3 (including its graphs). Better explain the use of coordinates (example: in equation (1) there appears a change from geodetic coordinates to navigation coordinates through a matrix multiplication but the coordinates are latitude, longitude and height. The navigation coordinates are not defined either and it seems that a displacement term is missing).

5-      The parameters of the WGS84 ellipsoid can be referred to a bibliographic reference and table 1 can be deleted.

6-      In section 5 indicate that traceless Kalman Filter is equivalent to Uncested Kalman Filter.

7-      In section 6 you can introduce some comments about the error in the simulation results due to the finite number of trials.

Comments on the Quality of English Language

Minor editing of English language required

Author Response

(The authors gave the same response as above.)

Round 2

Reviewer 1 Report

Comments and Suggestions for Authors

Thanks for addressing all my comments on your article, I think you did a great work to improve the manuscript. I especially appreciate the fact that you performed experiments to test your localisation algorithm. 

All in all, I do not have any particular comment. I find it a pity that you cannot share the source code after publication. 

Comments on the Quality of English Language

The language improved a lot, the text is clearer now.

I would still have another go with Grammarly or similar. You still have very long sentences and paragraphs.

You also need to increase the font size inside pictures. It is really difficult to read when printed..

After this, I think the paper is good for publication. 

Author Response

Dear Reviewer,

Thank you very much for your hard work during the review process of this article. In response to your feedback, we have carefully reviewed and improved it again, and marked the modified parts in the main text for your review. Please refer to the attachment for specific content.
